# Biomonitoring of Airborne Microplastic Deposition in Semi-Natural and Rural Sites Using the Moss *Hypnum cupressiforme*

**DOI:** 10.3390/plants12050977

**Published:** 2023-02-21

**Authors:** Fiore Capozzi, Maria Cristina Sorrentino, Eleonora Cascone, Mauro Iuliano, Gaetano De Tommaso, Angelo Granata, Simonetta Giordano, Valeria Spagnuolo

**Affiliations:** 1Dipartimento di Biologia, Università degli Studi di Napoli Federico II, Via Cupa Nuova Cintia, 21-80126 Napoli, Italy; 2Dipartimento di Scienze Chimiche, Università degli Studi di Napoli Federico II, Via Cupa Nuova Cintia, 21-80126 Napoli, Italy

**Keywords:** FT-IR microscope, atmospheric pollution, passive biomonitoring, plastic polymers, microfibers

## Abstract

We show that the native moss *Hypnum cupressiforme* can be used as a biomonitor of atmospheric microplastics (MPs). The moss was collected in seven semi-natural and rural sites in Campania (southern Italy) and was analyzed for the presence of MPs, according to standard protocols. Moss samples from all sites accumulated MPs, with fibers representing the largest fraction of plastic debris. Higher numbers of MPs and longer fibers were recorded in moss samples from sites closer to urbanized areas, likely as the results of a continuous flux from sources. The MP size class distribution showed that small size classes characterized sites having a lower level of MP deposition and a high altitude above sea level.

## 1. Introduction

The enormous potential for use of plastic has been clear since its invention [1]. Indeed, in a few decades plastic has invaded our lives; for example in packaging, synthetic textile fibers, disposable plastic accessories, vehicle components, and structural parts. There has been an exponential growth in plastic production in recent years, which has substantially overtaken any other manufactured materials [2]. The benefits of plastic in everyday use are recognized worldwide: versatility, practicality, resistance, and durability. This has ensured that plastic objects are preferred over similar materials that are less resistant and perishable. However, it is precisely because of this durability that has resulted in its role in environmental problems, since plastic objects are often used only once or for a short time, such as a bottle of water, take over several hundred years to degrade, with the consequent accumulation of plastic waste and contamination of all environmental matrices. For example, it is estimated that four trillion plastic bags are used every year and one million plastic water bottles are used every minute in the world [3].

Plastic fragments are typically categorized by size into macroplastics (>5 mm), large microplastics (1–5 mm), microplastic (1 μm to 1 mm), and nanoplastics (≤1 μm) [4]. Microplastics (MPs), include primary MPs, which are directly produced in form of small-sized particles, as micro- and nanobeads; and secondary MPs, deriving from degradation or fragmentation of large plastics [5]. According to their structure and shape, MPs can be classified as fibers, foams, beads, fragments (including film), and pellets [6]. About 10 years ago, MPs were initially detected in the oceans and marine organisms, but recently, an increasing number of studies have shown the presence of MPs in soil, freshwater, sediments, and air [7,8,9,10,11]. Recent studies estimate that plastic leakage is about 10 million tons per year, of which 3 million tons are primary MPs. According to Brahney et al. [12], around 1100 tons of microplastics are present in the atmosphere in western U.S.A. and Europe; sources such as road traffic, can alone produce about 1327 million tons of microparticles per year [13]. Furthermore, MPs such as polystyrene (PS) or polyvinyl chloride (PVC) can contain many inorganic and organic pollutants, which are known to be mutagenic and carcinogenic for organisms coming in contact with them, mainly by ingestion and inhalation [14].

Plastics can contribute to climate change since they mostly derive from fossil fuel and emit greenhouse gases at each stage of their life cycle [15]. It is predicted that plastic production will emit over 56 billion Mt of carbon dioxide equivalent in the next 30 years [16]. According to Ford et al. [17], plastics can contribute to climate change in three ways: (1) plastic production, transport, and use; (2) plastic disposal, mis-managed waste, and degradation; and (3) bio-based plastics, which are completely or partially made from biological resources.

There is also evidence of MPs in remote environments. Napper et al. [18] found plastic debris, mainly fibers, in stream water and snow collected from several locations on Mount Everest, including the Balcony at 8440 m.a.s.l. According to the authors, tourism could be the primary cause of MP contamination and its increase could even worsen the situation. Plastic has also been found in the wildlife of remote areas; for example, the Andean condors accumulate microplastics through their diet composed by plastic-contaminated carcasses of pinnipeds and South American camelids [19]. Taylor et al. [20] found that organisms living in the deep-sea floor with different feeding mechanisms (i.e., Cnidaria, Echinodermata, and Arthropoda) ingested and internalized microplastics. This indicates a worrying contamination by microplastics through the trophic chain, affecting animals living in potentially uncontaminated environments.

Therefore, the presence of MPs in the biosphere has become a global concern, also affecting the ecosystem and possibly human health. This makes it necessary to monitor MPs and track their fate in the environment as a first step to adopt efficient strategies to mitigate their proliferation.

Biomonitoring with either seed plants or cryptogams, thanks to its feasibility and relatively low costs, allows us to obtain large amounts of data in a short time frame [21,22,23,24,25]. Therefore, plant biomonitors often represent a viable and cost-effective option to moni tor MP deposition rates, with the advantage of multiple sampling points, thus providing a proper representation of pollution in a region compared to the more expensive non-biological methods. The biomonitoring of atmospheric MPs could also allow the implementation of depositional models and risk maps that are associated with their presence. Cryptogams are well-known as biomonitors of organic and inorganic pollutants, mostly those present in the atmosphere in the form of particulate matter; these plants have been used in both passive and active biomonitoring of air quality [26,27]. Recent studies with cryptogams, particularly mosses, have shown their potential for biomonitoring MPs with mosses intercepting MPs (polystyrene nanoparticles) in aqueous environments [28], and native moss species accumulating airborne MPs, even in sites far from urban centers [29]. One such moss species, *Hypnum cupressiforme* Hedw., is one of the most widely used biomonitors, due to its abundance and its unique capacity for accumulating environmental toxins [30]. However, studies investigating this species for MP biomonitoring are, to date, absent.

Campania is among the Italian regions with the highest number of inhabitants, resulting in an enormous production of waste and in recent years, this region has made headlines due to open-air dumping and the illegal incineration of industrial toxic waste [31,32]. Despite this evidence, the presence of airborne dispersed MPs in the Campania region is indeed poorly investigated in the literature; there are few studies on this topic, mainly concerning aquatic environments and evaluating the presence of these pollutants in rivers, lakes, or sea (e.g., [33,34,35]). Given its topographical and geographical complexity, i.e., presence of mountains, lakes, rivers, and a patchwork of urban, agricultural, and industrial areas and the high cost of management of the monitoring stations, Campania has a limited number of locations devoted to the monitoring of atmospheric pollution. Biomonitoring with cryptogams, thanks to its ease of application and relatively low cost, represents a good alternative for air quality assessment. However, this method needs to be tested to estimate the MP abundance in the region and above all, to identify the areas with lower depositions, which can be used as sources of plant material for future transplant experiments. The latter experimental approach offers a wide range of applications compared to the sole collection and analysis of native species [27,36,37,38].

We hypothesized that various types of MPs are present throughout Campania and that site characteristics, such as altitude, distance from urban centers and population density, could affect their abundance and local distribution. Moreover, we hypothesized that the native moss *H. cupressiforme* is capable of accumulating airborne MPs. To test these hypotheses, we investigated the abundance and dimensional class distribution of airborne MPs on the moss *H. cupressiforme* collected in seven semi-natural and rural sites of the Campania region.

## 2. Results

### 2.1. Number of MPs

Most of the observed debris belonged to the category of microfibers (99%); therefore, these have been counted and characterized and will be the subject of following discussions. However, we use the generic term ‘microplastics’ (MPs), unless specified differently. Microplastics were found in all samples at all collection sites (Figure 1). A total number of 1491 MPs were found across all replicas and sites, the average number of microplastics found in all sites was 71 ± 13 MPs g^−1^. MTB, MTV, and MTS sites were characterized by a lower average number (mean ± s.d.: 53 ± 4, 57 ± 9 and 65 ± 11 MPs gr^−1^, respectively). The collection site MTF showed the highest number 87 ± 7 MPs gr^−1^, although it was not statistically different from the ARP (80 ± 3 MPs gr^−1^), PTR (78 ± 10 MPs gr^−1^), and MRC (78 ± 4 MPs gr^−1^) sites.

### 2.2. MPs Dimensions and Distribution

The length of plastic debris across all sites ranged from 0.2 to 10.00 mm, but only those up to 5 mm (i.e., microplastics) were considered in the data analysis. The longest particle was observed at the MTF site (10 mm) followed by the MTV (8.3 mm) and the ARP (7.5 mm). The average MP length ranged from 1.12 mm at the MTB site to 1.68 mm at the MTF site (Figure 2).

Microplastics deposited on *H. cupressiforme* at the MTF site were significantly longer. MP distribution divided into size classes with a step of 0.2 mm from 0 to 5 mm (Appendix A; Figure 3), highlighted that moss samples from MTF, followed by PTR, had higher percentages of MPs falling into the larger dimensional classes, compared to the other sites. By contrast, MTB and MTV showed mainly shorter fibers, with percentage peaks over 10% for size classes 0.4–1 mm.

### 2.3. FT-IR Characterization

Thirty fibers visually identified as MPs were characterized with the Nicolet 5700 spectrometer connected to Fourier Transform infrared (FT-IR)-microscope (Appendix A). All the fibers were polymers; the most abundant was the polyethylene terephthalate (PET) ~75%, followed by polypropylene (PP) ~15%, and polystyrene (PS) ~8%, and the remaining 2% was composed mainly of polyvinyl chloride (PVC) (Figure 4).

## 3. Discussion

There are relatively few studies that explore the possibility of using moss as biomonitors of airborne MP [29,39] and, to our knowledge, this is the first contribution that is focused on the comparison between moss from different semi-natural and rural areas in Italy. Microplastics were found in *Hypnum cupressiforme* at all of the sites investigated, and in accordance with recent observations [40], we found a higher proportion of fibers. The highest number was found at MTF, the site closest to the large town, which is densely populated, Castellammare di Stabia and also Napoli and Salerno cities, followed by the rural sites in the Avellino and Benevento provinces (ARP, PTR, and MRC), which are located at a relatively lower altitude a.s.l. The ARP and PTR sites are rural areas, whereas MRC is near to a highway. Therefore, our observations support the hypothesis that the closer the anthropized areas, the higher the number of MPs deposited on the moss. Similarly, in a study on *Hylocomium splendens* (Hedw.) Schimp, collected at three natural sites from Ireland, Roblin, and Aherne [29] found on average 24 microfibers g^−1^ in dry moss, with increasing numbers at the sites closest to urban centers and a maximum of 34 microfibers g^−1^ in Glendalough, which is located <50 km from Dublin. In a study with lichen transplants conducted in Milan, Jafrova et al. [40] found 20–26 MP/g in the control site and urban parks, and a higher number of 44–56 MP/g in central and peripheral areas, indicating that the urban areas are the most affected by MP fallout.

It is also worth noting that the lowest number of MPs was observed in sites at higher altitudes above sea level, even in sites close to each other, but with different altitudes (i.e., MTV, near to MRC). This suggests that altitude is an important factor in the deposition of airborne MPs. It is likely that larger MP debris cannot reach higher altitudes due to their weight [29]. In addition, MPs detected in the present study were mostly fibers, which are lighter and foldable compared to other plastic debris, and were likely offering less resistance to resuspension and transport into the atmosphere.

The highest deposition of MPs in *H. cupressiforme* collected at MTF could also depend on the proximity to the sea: this site is located in the Sorrento peninsula, between the gulfs of Napoli and Salerno (Figure 4). This hypothesis is supported by the direction of prevailing winds, mainly blowing from south–west, and from the sea to the shore, and sea sprays are known to be vector for the diffusion of MPs in the atmosphere [41]. MP length distribution among the collection sites highlighted that the longest MPs were found at MTF, the site located between Napoli and Salerno urban areas. It is likely that the proximity to sources (i.e., urban and industrial areas, highways and trafficked roads) determined a continuous flux of plastic fragments, with a rate of deposition greater than degradation, and a consequent accumulation of larger fragments. In agreement with our results, Loppi et al. [39], studying along a transect the accumulation of airborne microplastics in lichens near a landfill dumping site, observed that the lichens closest to the landfill accumulated the highest concentration of MPs with the length decreasing with the distance from the landfill. Roblin and Aherne [29] found a higher proportion of shorter microfibers at sites that were more remote compared to urban centers, thus supporting the hypothesis that smaller fibers are more likely to be transported to remote sites. Similarly, Jafarova [40] found the longest fibers in the sites chosen at the city center and in the semi-periphery.

The plastics found on moss belong to the category of thermoplastic polymers, materials, which become soft or moldable at a high temperature (65 °C and 200 °C, respectively [42]). Similar to our findings, the same polymer proportions were recently observed by Abelouah et al. [43] in a study conducted in the Central Atlantic Ocean (CAO) of Morocco, further confirming their widespread diffusion in the environment. In a study analyzing wastewater, stormwater runoff, and surface water, Yano et al., [44] found PET to be dominant in road runoff, followed by PP and PE. Polyethylene terephthalate (PET) is the most widely used plastic in the world, it is made with petroleum, natural gas, or vegetative raw materials. It is completely reusable and retains its fundamental properties during the recycling process. Due to its chemical stability, PET packaging complies with the restrictions concerning hygienic standards imposed on the food, cosmetic, and pharmacological industries. Polypropylene (PP) is widely used as a film for packaging, in the textile sector, and for components of automobiles, and thanks to its chemical and physical resistance, it is also used in the production of components for the chemical industry. It is relatively cheap despite its high technical properties, and this facilitates its spread. Polystyrene (PS) is more brittle than PET and sensitive to petroleum and organic solvents. It is used to create disposable tableware and packaging. Polyvinyl chloride (PVC) is a plastic of higher consumption, it is used for pipes, electric cable covers, safety helmets, imitation leather, fixtures, etc. [45]. Consequently, the diffusion of these polymers, even in natural and rural sites, can be explained by their intensive and widespread use and improper disposal, which is linked to the high population density and large urban and industrial areas that characterize Campania.

Based on both the number and the length of MPs found on *H. cupressiforme,* MTB and MTV are the sites with the lowest level of depositions of MPs, and therefore, the most suitable for the collection of this moss to be used in active biomonitoring. However, we are aware of the potential limitations of the methodology applied for airborne MP estimation. First of all, the visual identification of the plastic fragments cuts out a part of MPs and all nano-plastics; secondly, the ability of the biomonitors to retain MPs is still poorly understood; and, thirdly, their chemical characterization is expensive and time-consuming. Indeed, the times and costs remain a concern even when considering the need for collecting and analyzing many samples from different sites to ensure a more accurate evaluation of the depositions over a large geographical area.

In general, the ability of *H. cupressiforme* to retain pollutants moving in the air being linked to particulate matter has been studied in depth. It is known that PM retention, mostly in the range of 1–10 µm [46], is essentially based on passive mechanisms [23]. Furthermore, it depends on the thallus structure, micromorphology [46,47,48], cell wall properties, and the chemical composition [30]. Further, it was assessed that once intercepted and retained, PM adsorption is quite stable, i.e., particle detachment hardly occurs [49]. However, the different chemical properties of MPs compared to other particulate matter (i.e., metals and metalloids, soil resuspension particles, and PAH linked to PM) require specific experiments to test the MP interception and retention ability by moss. An experiment carried out in a water environment using polystyrene nanoparticles indicted that they were adsorbed by the moss *Sphagnum palustre* L., even though washing removed larger particle aggregates [28]. Furthermore, MPs are non-polar molecules, which makes them poorly reactive and unstable in the bond with the moss tissue. To explain the mechanism of such a bond, we hypothesized the occurrence of weak interactions between benzene residuals of styrene and polar molecules of the cell wall [28], but of course, different relations could occur at the interface between moss and air. Interestingly, some previous data on metal(loid)s accumulated in the same species are in line with the data presented here. For example, Vingiani et al. [50] found that the element concentration in *Hypnum cupressiforme* collected at MTF was higher than the concentration measured in the same species collected at MTB.

## 4. Materials and Methods

### 4.1. Study Area and Sampling Sites

All the collection sites were selected in Campania (southern Italy). At national level, it is the third region for number of inhabitants (5,569,833) and the second for population density. It is spread over 13,670.95 km^2^ and it is located between the Tyrrhenian Sea (SW) and the southern Apennines (NE). The regional capital is Napoli (909,906 inhabitants), which is one of the most densely populated cities in Europe [51]. The prevailing winds in Campania region vary according to the season: in autumn and winter they are from the west or south-west, while in spring and summer they are from the north and north-east. [52].

The sampling sites were selected based on the presence of the target species, namely *Hypnum cupressiforme* Hedw., in semi-natural areas, S-N, (i.e., unaltered, near-natural areas in which the anthropic impact is negligible or virtually absent) and rural areas, R, (i.e., a countryside located outside towns and cities with small settlements devoted to agricultural and forestry activities (Table 1 and Figure 5)).

### 4.2. Sampling

The pleurocarpous moss *H. cupressiforme* was collected following the recommendation by ICP Vegetation [53]. The samples were collected by wearing nitrile gloves and cotton wears from a 50 m^2^ plot at each of the 7 sites (~10 g wet weight at each site); all the samples were collected away from any possible source of local contamination including trees, roads, and anthropogenic manufactures, and were placed in glass jars and then transported to the laboratory.

### 4.3. MPs Extraction

The moss samples were manually cleaned from extraneous materials, soil, and other plants debris under a stereomicroscope wearing nitrile gloves and cotton dresses, and were then dried at 50 °C for 48 h. Under a laminar hood, three replicas of 1 g moss, obtained mixing the 10 g moss samples for each site, were digested using the wet peroxide oxidation method described in Masura et al. [54] and Herrera et al. [55]. The digestion procedure was carried out by adding 40 mL of 0.05 M Fe (II) solution per gram of moss, and then 40 mL of 35% H_2_O_2_; the mixture was left at room temperature for 5 min. To boost the reaction, the digestate was heated to 55 °C, and further 20 mL aliquots of H_2_O_2_ were added when the reaction slowed down if organic matter was still visible. The samples were then vacuum-filtered onto glass-fiber filter circles (MN GF-4, retention capacity: 1.4 µm), following Dris et al. [11], and to better distinguish between synthetic material and organic matter, the samples were dyed with 1 mL of the biological stain Rose Bengal (4,5,6,7-tetrachloro-2′,4′,5′,7′-tetraiodofluorescein, 200 mg L^−1^), following Liebezeit et al. [56] and Kosuth et al. [57]. Finally, the moss sampling glass jars were rinsed with MilliQ water and filtered to capture residual fibers. The dyed filters were transferred to glass Petri dishes for storage and for assessment of MPs. Throughout sample collection, processing, and analysis, procedural open-air blanks were used to determine potential contamination. Water and peroxide blanks were vacuum-filtered and analyzed following the same method as for the moss samples to determine possible contamination by MPs.

### 4.4. Stereomicroscope Identification and FT-IR Analyses

The stereomicroscope (Leica Wild M8) was used for the MPs identification following the five visual identification criteria from Windsor et al. [58]. In the current study, to provide data comparable with previous studies [29] non-stained fibers that met at least two of the criteria were classified as anthropogenic MPs. The stereomicroscope was implemented with a micrometric grating lens for measuring the MPs. To unambiguously prove the synthetic nature of the non-stained MPs, 30 particles from all sites were randomly selected and were observed under a Nicolet 5700 spectrometer connected to Fourier Transform infrared (FT-IR)-microscope.

### 4.5. Data Analysis

All data were processed using Microsoft Excel and IBM SPSS Statistics for Windows (IBM Corp. Released 2020, Version 27.0. Armonk, NY, USA: IBM Corp). The normality and homogeneity of the variances of the datasets were assessed using the Shapiro–Wilk’s test and Levene’s test, respectively. The differences among means of different groups were assessed by ANOVA.

## 5. Conclusions

This first study focused on the deposition of MPs in semi-natural and rural sites of southern Italy and shows that the airborne pollution of MPs is widespread. We found that MP deposition decreased with the distance from highly populated urbanized areas and altitude. Moreover, fiber length was on average shorter at higher elevation sites. MTB and MTV are the sites with the lowest level of MPs, and therefore, the most suitable for the collection of *H. cupressiforme* to be used in active biomonitoring studies. In future, it is of primary importance to carry out similar studies to understand the mechanisms affecting the deposition of MPs on plants. For this purpose, the ability of cryptogams and leaves to retain MPs should be investigated in depth to highlight the structural traits that favor MP accumulation. Furthermore, transplanted cryptogams should be tested at different exposures and time conditions. Such studies would lead to the production of a standard protocol, which is indispensable for achieving a high methodological rigor and comparing the results obtained in different studies.

## Figures and Tables

**Figure 1 plants-12-00977-f001:**
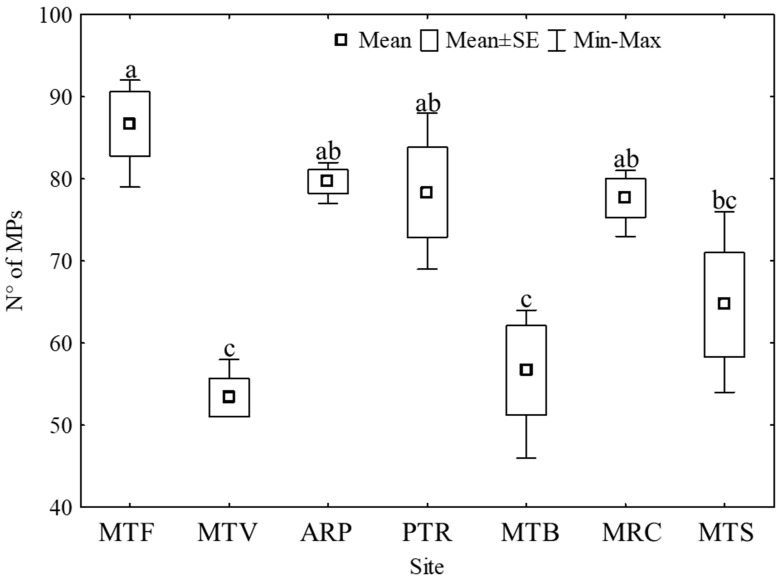
The number of microplastics (MPs) per gram of samples (d.w.) from the 7 collection sites. Different letters indicate significant differences according to Tukey’s test, *p* < 0.05.

**Figure 2 plants-12-00977-f002:**
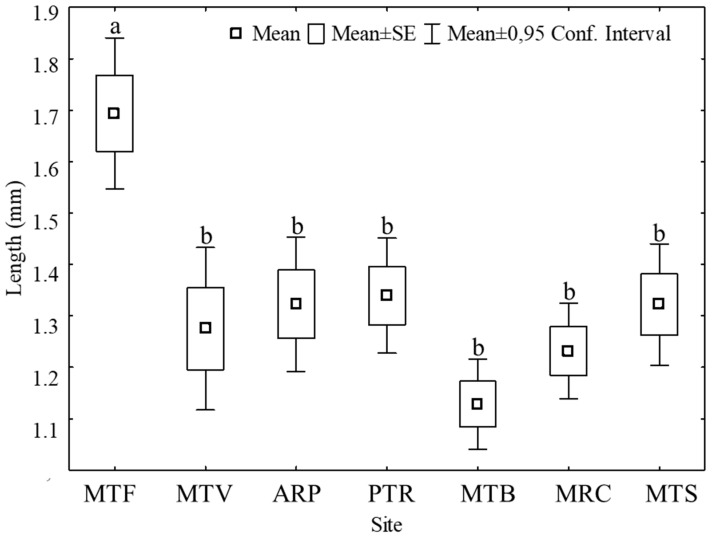
The length of microplastics (MPs) at the 7 collection sites. Different letters indicate significant differences according to Tukey’s test, *p* < 0.05.

**Figure 3 plants-12-00977-f003:**
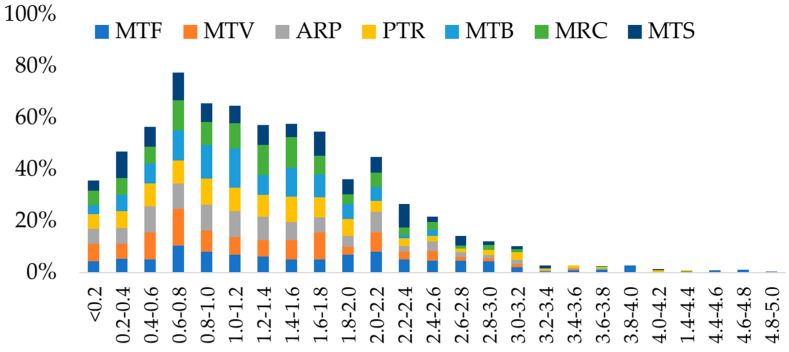
Microplastic distribution divided into size classes from 0 to 5 mm. In each dimensional class only the lower value is included.

**Figure 4 plants-12-00977-f004:**
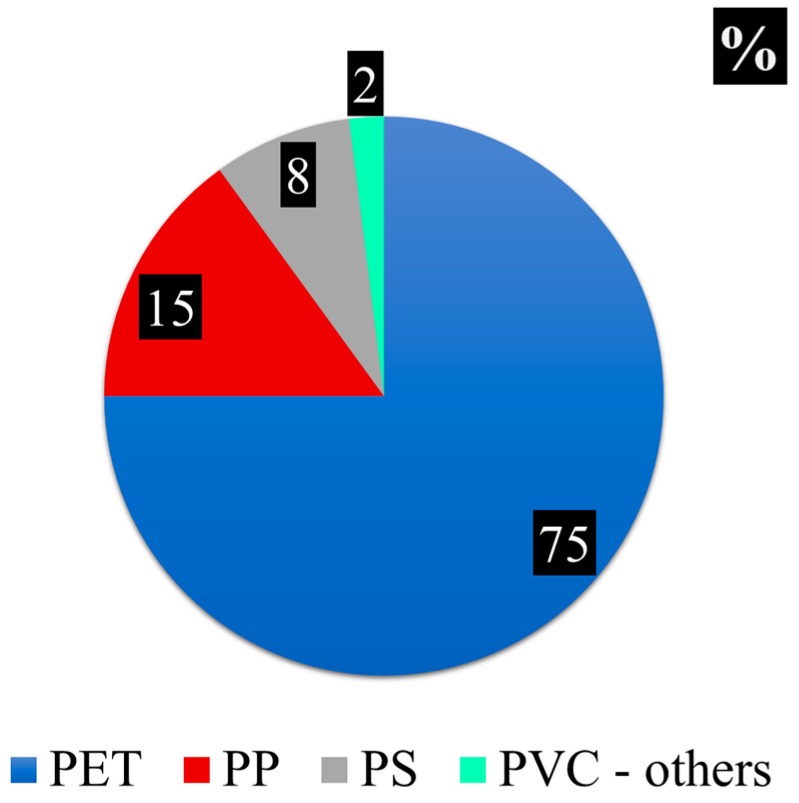
Results from FT-IR polymer characterization based on 30 observed spectra. Note—PET: polyethylene terephthalate; PP: polypropylene; PS: polystyrene; and PVC: polyvinyl chloride.

**Figure 5 plants-12-00977-f005:**
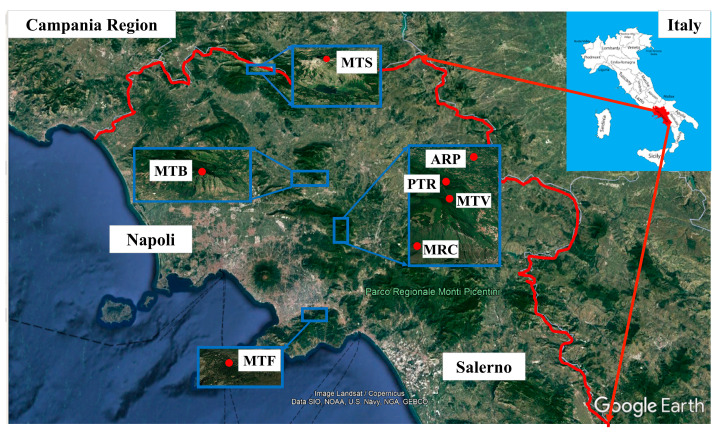
Campania region view from Google Earth^®^ and the seven sampling sites.

**Table 1 plants-12-00977-t001:** Information on sampling sites.

Site Name	Site Code	Lat.	Long.	Altitude m a.s.l.	Distance * (km)	Municipality	Inhabitants km^−2^	Site Type **
Monte Faito	MTF	40.669338°	14.479714°	1.044	1.7	Castellammare di Stabia	3526	S-N
Mercogliano	MRC	40.922883°	14.723577°	650	1.3	Mercogliano	580	R
Monte Vergine	MTV	40.962178°	14.718305°	1.255	3.4	Ospedaletto D’Alpinolo	364	S-N
Pietrastornina	PTR	40.986610°	14.710573°	894	1.5	Pietrastornina	91	R
Arpaise	ARP	41.029755°	14.752552°	510	1.0	Arpaise	109	R
Monte Taburno	MTB	41.102701°	14.603371°	1.050	3.5	Tocco Caudio	52	S-N
Monte Matese	MTS	41.468321°	14.405094°	1.300	1.3	San Massimo	30	S-N

* Distance from the closest urban settlement (municipality). ** S-N: semi-natural; R: rural. Sites were classified according to Corine land cover for Campania Region: https://sit2.regione.campania.it/node (accessed on 24 January 2023).

## Data Availability

All data are reported in the text and in the Appendix A.

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
