# Peer review of "Biomonitoring of Airborne Microplastic Deposition in Semi-Natural and Rural Sites Using the Moss Hypnum cupressiforme"

_plants, 2023, doi:10.3390/plants12050977_

Round 1
Reviewer 1 Report
plants-2054381
Biomonitoring of airborne microplastics deposition in proximal-natural and rural sites using the moss Hypnum cupressiforme (Hedw.)
Capozzi Fiore and colleagues
The manuscript presents data on microplastics deposition (assessed through moss biomonitoring) from the Campania region in southern Italy. This study adds to the growing number of studies (albeit still limited) that have evaluated moss and lichen as biomonitors of atmospheric deposition. I have very few comments on the manuscript and recommend that it is accepted for publication. However, the written English could be greatly improved; while the manuscript is understandable, the science is obscured in places. I believe this is a good contribution to the literature; it could be a great contribution with an improvement in the written English.
L42. Ensure consistent use of spacing between value and unit, 1–5 mm, 1 µm, etc.
L58. There is growing concern on the potential effects of microplastics but there is limited (no) evidence of ecosystems and humans being ‘deeply’ affected.
L63 to L73. The authors are missing relevant background here. There are already several studies that have used plants (mosses and lichens) as biomonitors of microplastic deposition. These studies should be introduced here.
L78 and throughout. The written text could be much improved; perhaps the authors could as a colleague that is a native English speaker. Words such are ‘individuate’ are understandable but perhaps the authors mean ‘identify’?
L83 to 85. Delete.
L91. What was the total count of microplastics across all sites? What was the site average? Add this information.
L98. Please clarify, is this per gram dry weight?
L98. It is unclear how statistics were applied to the data as they are only counts. Secondly, how did the authors decide that parametric statistics were appropriate?
L101. Dimensions > 5 mm indicate macroplastic. The aim of the study (L77) is focused on microplastic.
L103. Does the average microplastic length include marcoplastics?
L103. Use past tense (throughout)
L120. What fraction (proportion) were identified? Where they all fibres?
L159. Please clarify (confirm): was fibre length shorter at higher elevation sites? Did you see fragments at urban sites?
L178. Number or number concentration?
L221. The term ‘proximal nature areas’ is mentioned several times (abstract and introduction) but it is unclear what this means. Proximal to what?
L264. Please confirm, were all 30 particles plastic polymers?
Author Response
Please see in the attachment

Reviewer 2 Report
After reading the manuscript, I have to say the manuscript cannot be accepted for publication in its present form.
My first comment is related to the manuscript structure. The authors should use the most prominent norm for the structure of the manuscript.
From my experience and knowledge, the authors cannot make any correct conclusions from such a low number of observations. Consequently, the discussion is very poor because the quality of data presentation and data interpretation is very low. The statements need to be supported with more qualitative and quantitative data.
The graphical work needs to be more informative and understandable for the readers. I am afraid that the manuscript cannot be improved with current conceptualisation and visualisation, thus I recommend thinking in this direction and substantially improve the manuscript prior its resubmission.
I regret to inform you that the manuscript cannot be accepted for publication in Plants.
Reviewer 3 Report
The manuscript is original and possesses a scientific potential in the field of biomonitoring of MPs. Nevertheless, some flaws have been identified and must be corrected before the acceptance of the article. The main flaw is that the manuscript lacks consistency. In general, the authors should have chosen a background area first, and not subsequently presented it both as a task and as an outcome. A hypothesis is also missing.
Discussion is relatively weak. Although there is scarce data, a comparison can be made with the data reported by Roblin and Aherne for Hylocomium splendens from background regions in Ireland. There are also some data for MPs in lichens in the literature. The conclusion does not include findings and implications.Some additional comments were included in the attached pdf.

Round 2
Reviewer 3 Report
Unfortunately, I only see a minor and stylistic correction in the declared objectives - Line 77-81. There is not a single new reference added to the discussion. I would like to emphasize that what the Authors quote about the conclusion applies as well as in the quote itself: “…segue into the future directions your study might inspire”. In other words - based on your findings, what remains as a need for further studies, not what you personally plan. In conclusion, I find no improvement in the manuscript, and for this reason I believe that the Authors should reconsider it.
Round 3
Reviewer 3 Report
Based on the second revised version and cover letter, I think the manuscript could be published.
Author Response
- The overall quality of the text should be improved and would require a deep check by a mother-tongue colleague.
àWe agree with the editor that some points in the text were cryptic and required redrafting and proofreading by a native speaker. Therefore, we thank the editor who gave us the opportunity to better explain these points and further improve our manuscript. The text has been revised by a native speaker colleague who was acknowledged, Prof. K.J Duffy.
- There are some critical points which should be still overtaken before accepting the paper. - The sampling design is not suitable to identify background sites to collect moss for active biomonitoring of MPs, as indicated in L217. It should have been based on a prior characterization of sites (including natural remote sites, which have not been selected). This point should be reformulated and modified throughout the manuscript. Anyway, the identification of background sites mentioned in introduction has not been reported at all in the conclusion.
à As now specified in the text, our aim was to verify the widespread presence of MPs in semi-natural and rural areas of Campania region. At the end of the present MS we suggest the less impacted sites as useful for moss collection in future transplant experiments. In this respect, it is important to point out that to date we do not have any other information about MPs content in native moss species of our region; this information is useful for the application of the technique since a clean thallus (clean from microplastic or in general from pollutants) improve the sensitivity of the methodology (Capozzi et al, 2017 - Environmental Pollution 225 (2017) 323-328). However, we recognize that the identification of "background-sites" must be associated with a different experimental design, therefore, we mitigated the concept and reformulated related sentences by deleting the term “background”. In the present version of the MS we only indicated the sites having a lower MP fallout (i.e., MTV and MTB).
- There is not a clear hypothesis tested in the manuscript.
à In the first round of review, we added the assumptions statement as requested by REV3; consequently. in the second round of revision the same REV3 replied "Based on the second revised version and cover letter, I think the manuscript could be published." Nevertheless, in the present MS we reformulated our sentence: We hypothesized that various types of MPs are present throughout Campania and that site characteristics, such as altitude, distance from urban centers and population density, could affect their abundance and local distribution. Moreover, we hypothesized that the native moss H. cupressiforme is capable to accumulate airborne MPs. To test these hypotheses, we investigated the abundance and dimensional class distribution of airborne MPs on the moss H. cupressiforme collected in seven semi-natural and rural sites of Campania region.”
- At the end of introduction, it is indicated that “The results are discussed in relation to sites characteristics, and biomonitoring data collected in Campania during the last decades”, however there is no specific comparison with pollution data for the experimental sites and only generic reference to pollution in the region and on two of the sites.
à We agree with the Editor; we deleted this sentence in the Introduction. Also, we mitigated the relative comment in the Discussion. Unfortunately, as stated also in the Introduction, Campania suffers of a limited number of monitoring station, mainly embedded in the urban context of Napoli. Evidence of this situation can be found, for example, on the website of the Regional Environmental Protection Agency, at the site: https://www.arpacampania.it/rete-regionale (last check 19/01/2023). The few functioning monitoring stations collect data mainly on numerosity and not composition of PM2.5 and PM10. So that most data on metals and hydrocarbons deposition in this region were only provided by biomonitoring research.
- The meaning of the abbreviation of the localities is not reported. Such localities are partially described only in discussion and therefore the reading of the text is difficult. How such localities have been classified as rural / natural is not reported and not supported by references or by other descriptive information. Authors should have reported e.g., distance from the closest pollution sources, type of source (when they mention industrial settlements), concrete data of pollution (e.g., depositions) for each site.
à We added the name of each sampling site and the distance from the closest urban settlement. Sites were classified according to Corine Land Cover for Campania Region: https://sit2.regione.campania.it/node (last check, 24/01/2023).
As explained in the previous response, data about deposition are not available in the examined sites, neither in their proximity. We also added the distance from the closest urban settlement and municipality population in the Table 1. The sites were chosen in rural and semi-natural areas with a low anthropic pressure. In the absence of direct known MP pollution sources, like plastic factories or plastic storage areas, and industries, we considered the closest urban settlements as the nearest putative MP source. For some sites located near larger cities (e.g., MTB, between Napoli and Salerno centers), we considered these sources in the discussion. We changed the discussion accordingly to avoid misinterpretation.
- L510-514. The level of heavy metal depositions, used here as a proxy of the pollution level of the site, should have been compared to microplastics (size, type, number). A deeper work (e.g., correlations between HMs and MPs) should have been done to support such statements related to MPs.
àThe comparison with other pollutants was outside the scope of this work. In our discussions we have mentioned the data collected in previous work on metals as, they were also found to be relatively low at the site MTB and abundant at MTF, in line with MP data. Since the data on elements are limited, we mitigated the text in line 246-252.
- Affirming that MTB is a suitable background area is not supported by e.g, microplastic number, which seems in fact lower in site MTV. So why MTB and not MTV has been proposed as background area? Do the values of HM and MPs in MTB really correspond to background values? I think that the Authors should evaluate these points.
àAs previously answered, we deleted the term “background”. In the present version of the MS we only indicated the sites having a lower MP fallout among those investigated (i.e., MTV and MTB).
- L547. How do samples have been cleaned from extraneous materials? Which strategies have been adopted to prevent lab contamination with microplastics during sample handling. Has this point been assessed? Please, include these aspects in the methodological section.
àIn M&M it was already specified that: “Throughout sample collection, processing and analysis, procedural open-air blanks were used to determine potential contamination. Water and peroxide blanks were vacuum filtered and analyzed following the same method as for the moss samples to determine possible contamination by MPs.” In the present version, we also added the following sentence: “The moss samples were manually cleaned from extraneous materials, soil and other plants debris under a stereomicroscope wearing nitrile gloves and cotton dresses, then dried at 50°C for 48 h.”
- The quality of graphical material and overall presentation should be improved, to be suitable for Plants.
-During the revision process, we changed Figure 3, 4 and 5 according to reviewers’ suggestions. For the Figure 1 and 2 we did not receive any comments and we don't want to be wrong in changing their presentation. So, if the editor has particular requirements, we kindly ask him/her to give us more specific indication.
